# A Prospective Randomized Trial of Postural Changes vs Passive Supine Lying during the Second Stage of Labor under Epidural Analgesia

**DOI:** 10.3390/medsci5010005

**Published:** 2017-03-08

**Authors:** María Simarro, José Angel Espinosa, Cecilia Salinas, Ricardo Ojea, Paloma Salvadores, Carolina Walker, José Schneider

**Affiliations:** 1Department of Obstetricia y Ginecología, Hospital Universitario Quirón, Pozuelo de Alarcón, 28223 Madrid, Spain; mariasigon@yahoo.es (M.S.); jaespibar@yahoo.es (J.A.E.); cecisali@hotmail.com (C.S.); rojea.mad@quiron.es (R.O.); cwalker@quiron.es (C.W.); 2Escuela Universitaria de Enfermería, Universidad de Cantabria, 39001 Santander, Spain; paloma.salvadores@urjc.es; 3Facultad de Ciencias de la Salud, Universidad Rey Juan Carlos, Avenida de Atenas SN, 28922 Alcorcón, Madrid, Spain

**Keywords:** labor, postural changes, second stage

## Abstract

There exist very few studies comparing different postures or postural changes during labor in parturients with epidural analgesia. Aim: To disclose whether the intervention of a multidisciplinary nursing team including a physiotherapist during the second stage of labor improves the obstetric outcome in parturients with epidural analgesia. Design: Prospective randomized trial. Setting: University-affiliated hospital. Population: Women undergoing labor with epidural analgesia after a normal gestation. Methods: 150 women were randomized either to actively perform predefined postural changes during the passive phase of the second stage of labor under the guidance of the attending physiotherapist (study group), or to carry out the whole second stage of labor lying in the traditional supine position (control group). Results: There were significantly more eutocic deliveries (*p* = 0.005) and, conversely, significantly less instrumental deliveries (*p* < 0.05) and cesarean sections (*p* < 0.05) in the study group. The total duration of the second stage of labor was significantly shorter (*p* < 0.01) in the study group. This was at the expense of the passive phase of the second stage of labor (*p* < 0.01). Significantly less episiotomies were performed in the study group (31.2% vs 17.8%, *p* < 0.05). Conclusion: The intervention of a physiotherapist during the second stage of labor significantly improved the obstetric outcome.

## 1. Introduction

Since the advent and generalized use of epidural analgesia in the obstetric wards of developed countries, pain management during labor is no longer an issue. However, effective epidural analgesia carries with it an increased risk of changes in fetal position [1], prolonged labor [2] excessive use of oxytocin [3] and a higher rate of instrumental deliveries [4]. Despite recent improvements in schedules and timing of its administration, epidural analgesia is still often associated with a relative increase in the rate of cesarean sections [5] and of third- and fourth-degree perineal tears [6].

On the other hand, a recent Cochrane Review [7] showed that upright and ambulant positions during the first stage of labor were associated with a shorter duration thereof, as well as with a significantly reduced rate of cesarean sections. However, this review included mothers having received epidural analgesia, together with those not having done so. Indeed, one of the results of the meta-analysis was that ambulant positions were significantly associated with less likelihood of having received epidural analgesia.

To the best of our knowledge, this is the first randomized trial comparing postural changes during the second stage of labor with the standard passive supine posture, in which (a) all mothers in either group received epidural analgesia; and (b) all mothers in the study group adopting postural changes were constantly monitored by a multidisciplinary nursing team including a physiotherapist.

## 2. Materials and Methods

The present study constituted the nucleus of the PhD Thesis of the first author (M.S.), a physiotherapist, for which reason her presence as part of the attending multidisciplinary obstetric team was mandatory during all labors included into it. This was possible twice a week along the 17 months during which the study took place (from 1 August 2010 through to 31 December 2011). The study was carried out in a single private, University-affiliated center (Hospital Universitario Quirón, Madrid, Spain), attending approximately 1000 births per year.

Only parturients at term (between the 37th and 42nd week of gestation), after single low-risk pregnancies in vertex presentation, entering labor spontaneously, were included into the study. All received epidural analgesia during the first stage of labor, which was continued until after the birth was completed.

Exclusion criteria were previous cesarean section, induced labor, hypertensive disorders of pregnancy, intrauterine growth retardation and difficulties in understanding the instructions of the physiotherapist.

All women accepting to enter the study signed an informed consent form. The study itself underwent approval by the Ethics Committee of the reference hospital for all the hospitals of the health area, Hospital Universitario Puerta de Hierro, Madrid, Spain (ethical approval code HUPH: P.I.: 30/09).

During the study period, 150 women accepted to take part in it. They were randomly assigned either to the experimental group, in which the parturients were encouraged to actively perform postural changes during the second stage of labor, after full cervical dilatation, under the guidance of the attending physiotherapist, or to the control group, in which the second stage of labor took place throughout with the woman lying in the traditional supine position.

The positions adopted by the parturients in the experimental group were (1) sitting, with the back against a birthing ball; (2) kneeling, sitting on her heels, with the back held straight against the bed’s head, and the arms resting on its edge; (3) hands-and-knees, resting the chest against the birthing ball; (4) lateral decubitus, either with the lower leg flexed and the upper one stretched, or both flexed; (5) supine, either with the legs flexed or stretched. Each position was kept for a minimum of 5 min and a maximum of 30 min. The number of positions adopted varied according to the parturient’s choice and the individual duration of the second period of labor. It ranged from one position (4%), to adopting two (28%), three (48%) or four (20%) different ones throughout the second stage of labor. None adopted all five positions proposed. The favored position was hands-and-knees, followed by sitting with the back against a birthing ball, lateral decubitus, kneeling, and finally supine lying.

The number of postural changes undergone by the parturients varied between only one and seven (Table 1).

Women in both groups were encouraged to prolong the passive phase of the second stage of labor, before actively bearing down, until as late as possible: when the fetal scalp was visible through the vulva between contractions; when they felt an irresistible urge to bear down, despite effective epidural analgesia; when the established time limit for the duration of the passive phase of the second stage of labor according to the local protocol was overstepped (1 h) or, alternatively, when fetal monitoring indicated it due to suspected loss of fetal well-being.

The attending midwives evaluated the descent and rotation of the infant’s head during the whole procedure, which in its turn was supervised by the obstetrician on duty, in case complications arose.

Both groups were well balanced with regards to age, body mass index, weight gained during the gestation, gestational week at labor and parity, with no significant differences between these items (Table 2).

### Statistics

Qualitative variables were compared by means of contingency tables and Fisher’s exact test. Quantitative variables were expressed by their mean and its standard deviation, and compared by means of Student’s *t*-test. Results were considered significant when the corresponding *p*-value was <0.05. All data were processed by means of the SPSS statistical package (IBM Statistics, Armonk, NY, U.S.A.)

## 3. Results

There were significantly more eutocic deliveries (*p* = 0.005) and, conversely, significantly less instrumental deliveries (*p* < 0.05) and cesarean sections (*p* < 0.05) in the experimental group (Table 3).

The total duration of the second stage of labor was significantly shorter (*p* < 0.01) in the experimental group. This was at the expense of the passive phase of the second stage of labor (*p* < 0.01). No differences were registered in the active phase, from the moment at which the parturients began actively bearing down (Table 4).

The perineal outcome was significantly different between groups: episiotomies were significantly more frequent in the control group (31.2% vs 17.8%, *p* < 0.05), whereas first-degree perineal tears were significantly more frequent in the experimental group (55.7% vs 32.9%, *p* < 0.05). There were no significant differences between groups in the incidence of second- and third-degree perineal tears, although it must be noted that none was registered in the experimental group, versus five in the control group, which required suture of the anal sphincter. Nevertheless, all participants in the study were interviewed two years after delivery, and none in either group had symptomatic fecal or urinary incontinence needing treatment.

Despite the aforementioned differences, the perinatal outcome was uniformly good. Infant weights were not significantly different, and the only significant, albeit clinically irrelevant, difference registered, was in the first-minute Apgar score, which was significantly higher in the experimental group, but always within normal limits for both groups (8.38 ± 1.08 vs 8.81 ± 0.86, *p* < 0.05).

## 4. Discussion

Previous studies have already shown benefits associated with postures different from the traditional supine one during the second stage of labor. Bodner-Adler et al. [8], in a case control study involving 307 women, found that an upright, squatting position during the second stage of labor was significantly associated with less use of analgesia and oxytocin, and a lower incidence of episiotomies, if compared to the traditional supine position. There are, however, very few studies comparing postures alternative to the traditional supine one in parturients undergoing epidural analgesia, because movement is to some extent impaired in them, and must be actively encouraged. There are even fewer prospective randomized studies in this population.

Downe et al. [9], following a randomized prospective trial on the effect of position in the passive second stage of labor under epidural analgesia, reported that women randomized to a lateral position had a better chance of spontaneous vaginal delivery. Walker et al. [10] conducted a very similar randomized prospective trial, and concluded that delayed pushing and a modified lateral position during the active phase of the second stage of labor were significantly associated with less instrumental deliveries and less perineal trauma. In this study, in which one of us (C.W.) participated, postural changes during the passive stage of the second stage of labor were also encouraged, but in an unsystematic way; the posture adopted, if any, and its duration, left to the personal choice of the parturient. The fundamental difference with the present study is that the type of possible postural changes and their duration was predefined and constantly supervised by a physiotherapist, and that all of the parturients in the experimental group underwent at least one postural change for the prescribed length of time.

## 5. Conclusions

In conclusion, our study shows that the systematic adoption of defined postural changes for a defined length of time under the supervision of a physiotherapist during the passive phase of the second stage of labor is significantly associated with a shorter second stage of labor, less instrumental deliveries and cesarean sections and a better perineal outcome (significantly less episiotomies and no third-degree perineal tears in the experimental group). All these benefits, interestingly, were at the expense of improvements during the passive phase of the second stage of labor, i.e., the time lapse between full cervical dilatation and active bearing down, which our parturients were encouraged to hold back for as long as possible (“delayed pushing”). In fact, there were no differences between groups in the duration of the active phase, i.e., after beginning to actively push, or in the posture during this phase, which was conducted in all cases in the traditional supine lying position on a birthing table, with the legs held in stirrups.

## Figures and Tables

**Table 1 medsci-05-00005-t001:** Number of postural changes undergone by the parturients in the experimental group (*n* = 73).

Number of Postural Changes	Number of Women (%)
1	3 (4.1%)
2	6 (8.2%)
3	11 (15.1%)
4	29 (39.7%)
5	14 (19.2%)
6	9 (12.3%)
7	1 (1.4%)

**Table 2 medsci-05-00005-t002:** Obstetrical features of parturients in the study and control groups.

	Control Group (*n* = 77)	Experimental Group (*n* = 73)
**Age (years)**	33.9 ± 3	33.4 ± 3
**BMI (Kg/m^2^)**	27.0 ± 3.3	26.9 ± 3.2
**Weight gain (Kg)**	12.8 ± 3.7	12.4 ± 3.7
**Gestational week**	39.3 ± 3.7	39.5 ± 1.2
**Parity**		
**1st**	50 (64.9%)	48 (65.8%)
**2nd**	24 (31.2%)	23 (31.5%)
**>2**	3 (3.9%)	2 (2.7%)

BMI: body mass index

**Table 3 medsci-05-00005-t003:** Differences in mode of delivery between groups.

Delivery Mode	Control Group (*n* = 77)	Experimental Group (*n* = 78)	*p*-Value
**Eutocic**	39 (50.6%)	54 (74.0%)	<0.05
**Instrumental**	30 (39.0%)	18 (24.0%)	<0.05
**Forceps**	4 (5.2%)	1 (1.4%)	ns
**Vacuum**	22 (28.6%)	17 (23.3%)	ns
**Thierry’s spatulas**	4 (5.2%)	0	
**Cesarean**	8 (10.4%)	1 (1.4%)	<0.05

ns: not significant

**Table 4 medsci-05-00005-t004:** Differences in the duration of the second stage of labor between groups.

Duration (min)	Control Group (*n* = 77)	Experimental Group (*n* = 78)	*p*-Value
**Whole second stage**	124.30 ± 44.83	94.66 ± 32.78	<0.001
**Passive phase**	73.87 ± 33.59	50.77 ± 20.54	<0.001
**Active phase**	50.43 ± 24.56	43.89 ± 23.78	ns

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
