# Peer review of "A Prospective Randomized Trial of Postural Changes vs Passive Supine Lying during the Second Stage of Labor under Epidural Analgesia"

_medsci, 2017, doi:10.3390/medsci5010005_

Reviewer 1 Report

This study presents a simple research question with clear answers: the design is appropriate, sample size adequate, and the conclusions are sound as to impact clinical practice

Author Response

I can only thank the reviewer for his/her comments, since he/she does not demand any particular modifications of the manuscript

Reviewer 2 Report

This is a highly interesting study by a multidisciplinary team. This, however, renders it difficult to be accepted by either a gynecologic journal, the first author being a physiotherapist, or, conversely, by a physiotherapy or rehabilitation journal, most other authors being either gynecologists or nurses.

In this sense, since the present journal has a more general audience, it would probably be ideal for the scope of the article. As has been said, the study is innovative, interesting, and presents results that are potentially applicable in the clinic. I have no particular comments regarding the methodology employed, since it is a prospective randomized trial which has been apparently carried out in a neat way. 

The only additional information I would like to have, if it is anyhow possible to obtain it, are the long-term results regarding the urological or proctological outcomes in either study group. A relatively long time has passed since completion of the study, and it would be highly interesting to know if either approach resulted in more or less cases of urinary or fecal incontinence in the long run. 

If this information is not retrievable because of loss to follow-up of the involved patients, I would nevertheless recommend the paper for publication, because the results, as they stand, are interesting enough

Author Response

The reviewer´s comments are acknowledged with thanks. 

Concerning his/her remark about possible differences in either urinary or fecal incontinence after protracted follow-up, the participants in the study were indeed interviewed to control this aspect two years after delivery, and no differences were found between either group. Indeed, none of the parturients developed urinary incontinence that needed treatment of any kind. These data have now been included into the results section (highlighted in blue)

Academic Editor Report 

This article is a good piece of research. Methodologically it is correct and shows an excelent way of directing multidisciplinary research. The results are very enlightening since the C-section in the area of research is quite elevated and keeps rising.